# Good Reads and Easy Novels
# Readability and Literary Quality in a Corpus of US-published Fiction

**Yuri Bizzoni**
Center for Humanities Computing
Aarhus University, Denmark
`yuri.bizzoni@cc.au.dk`

**Pascale Feldkamp Moreira**
School of Communication and Culture
Aarhus University, Denmark
`pascale.moreira@cc.au.dk`

**Nicole Dwenger**
Aarhus, University, Denmark
`01805351@post.au.dk`

**Ida Marie S. Lassen**
Center for Humanities Computing
Aarhus University, Denmark
`idamarie@cas.au.dk`

**Kristoffer L. Nielbo**
Center for Humanities Computing
Aarhus University, Denmark
`kln@cas.au.dk`

**Mads Rosendahl Thomsen**
School of Communication and Culture
Aarhus University, Denmark
`madsrt@cc.au.dk`

## Abstract

In this paper, we explore the extent to which readability contributes to the perception of literary quality as defined by two categories of variables: expert-based (e.g., Pulitzer Prize, National Book Award) and crowd-based (e.g., GoodReads, WorldCat). Based on a large corpus of modern and contemporary fiction in English, we examine the correlation of a text's readability with its perceived literary quality, also assessing readability measures against simpler stylometric features. Our results show that readability generally correlates with popularity as measured through open platforms such as GoodReads and WorldCat but has an inverse relation with three prestigious literary awards. This points to a distinction between crowd- and expert-based judgments of literary style, as well as to a discrimination between fame and appreciation in the reception of a book.

## 1 Introduction and Related Works

Is it overall better for a novel to strive for an easy prose, or is there a link between difficulty and literary quality? The concept of readability has been studied for decades and is defined as the ease with which a text can be read and understood (Dale and Chall, 1949). Several works have attempted to define an easy way to compute readability in order to make, for example, didactic books more accessible, reduce technical jargon in documents produced for the general public, and adjust text selections according to the intended audience (Dubay, 2004). The result has been a series of popular and amply tested measures, each with a slight difference in their model of readability. Dale and Chall (1949), for example, referred to readability as the combination of elements in a text that impact important aspects of a reader's experience - including whether the reader can understand the text, finds it interesting, and can read with optimal speed (Dale and Chall, 1949). Despite their shortcomings (Redish, 2000), readability measures have been broadly applied to a large number of different domains. Measures of readability vary according to what aspect of a text they take into account, but they typically combine features such as sentence length, word length, and the presence of complex words. While the actual ease of a text depends on reader characteristics (background, situation, ability) it is widely accepted that simple textual features such as sentence length, syllables per word and lexical diversity impact the reading experience (Dubay, 2004). The connection of readability to the quality of a text has often been implied when it comes to non-fiction, and early studies into readability attest to the educational and social importance of developing such measures to improve technical or expository documents (Chall, 1947), but its role in the quality of *literary* fiction is much more complex. An easy-to-read novel can be enjoyable to read, but may also apppear poor or uno-

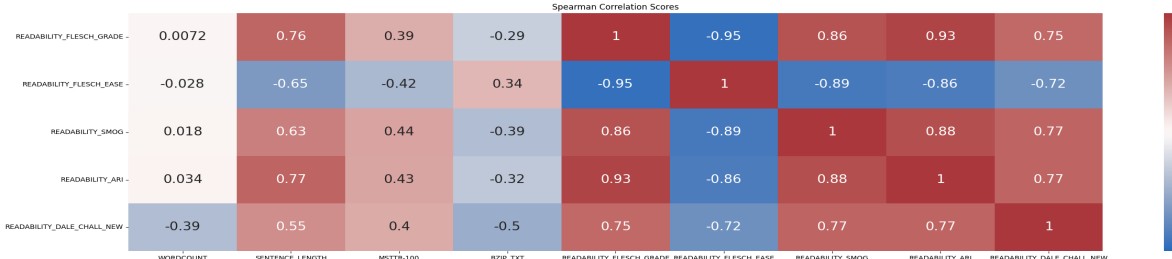

Figure 1: Correlations between stylometrics and flavours of readability (Spearman). All correlations between 0.09 and 0.99 are statistically significant.

riginal. In literary studies, the idea that readability might be a precondition for literary success is debated, and literary texts have been assessed variously by readability measures and similar metrics. Sherman (1893) was one of the first scholars to propose certain values of average sentence-length and reading ease as properties of "better" literary style. Readability naturally varies across genre, but it is a widespread conception for readers and publishers alike that bestsellers (as defined by top book-sales) are easier to read (Martin, 1996). More recently, readability has gained traction in areas of (commercial) creative writing and publishing, especially where its measures are implemented in text-editing tools such as the Hemingway or Marlowe editors [1]. These applications tend to favour lower readability scores - which is, texts easier to read. Yet, on the large scale, few studies have included readability as a measure that could help predicting literary quality. Studying a small corpus of bestsellers and more literary, canonical works, Martin (1996) found no significant difference in readability, using a modified Flesch reading score, while Garthwaite (2014) found differences in readability between bestsellers and commercially endorsed book-list titles. Relying on multiple measures of readability and one measure of literary quality (i.e., GoodReads' average ratings), Maharjan et al. (2017) found that readability was actually a weak measure for estimating popularity in comparison to, for example, character n-grams. Still, many studies of literary success, popularity, or perceived literary quality have sought to approximate text complexity and have studied textual properties upon which formulae of readability are directly or indirectly based, such as sentence-length, vocabulary richness, or text compressibility (Brottrager et al., 2022; van Cranenburgh and

Bod, 2017; Crosbie et al., 2013).

The question of the role of readability in literary quality is complicated by the practical and conceptual problem of defining literary quality itself, and consequently of quantifying it for large scale studies. Studies that seek to predict perceived literary quality from textual features often rely on the provisional proxy of one single gold standard, such as book-ratings from large user-platforms like GoodReads (Maharjan et al., 2018), personally or institutionally compiled canons (Mohseni et al., 2022) or sales-numbers (Wang et al., 2019). However, it has been shown that readers may have different, distinct perceptions of quality that are not necessarily based on the same criteria or prompted by the same textual features (Koolen et al., 2020).

In this paper, we explore to what extent readability might contribute to the perception of literary quality – defined through several alternative measures – in a large fiction corpus of modern and contemporary novels in English, taking into account, instead of one golden standard, different contextual perspectives on literary quality, so as to cover both crowd-based and "expert"-based standards of judgment.

## 2 Data and Methods

The essence of our approach consists in examining whether readability, as measured through five different algorithms, and literary quality, as approximated through six different resources, show any correlation on a large corpus of English-language fiction. We use standard correlation measures (Pearson and Spearman product-moment correlation coefficients $r_p$ and $r_s$, respectively). For inference on the correlation measures, simple Student's t-tests are used. For robustness checks, correlation coefficients were also modelled using a Bayesian ridge model of standardized the variables – al-

---

[1]https://hemingwayapp.com/help.html, https://authors.ai/marlowe/

though not reported due to limited space.[2]

## 2.1 Corpus

We use a corpus of modern and contemporary fiction in English, the so-called Chicago Corpus. [3] The Chicago Corpus is a collection of over 9000 novels from 1880 to 2000, representing works of fiction that are widespread in libraries, that is, the works of fiction that have a large number of library holdings as listed on WorldCat, a large-scale, international online library catalogue [4]. The number of holdings was used as a first filtering measure to include or exclude works in the dataset, yet there are still large differences in how many libraries hold each title, so we can use it as a metric to score different titles within the dataset as well. The corpus is unique, to our knowledge, for its diversity and extraordinary representation of famous popular- and genre-fiction, as well as seminal works from the whole period: key works of modernism and postmodernism as well as Nobel laureates and winners of major literary award. Still, it should be noted that the Chicago corpus reflects a clear cultural and geographical tilt, with a strong over-representation of Anglophone authors, and features only works either written in or translated into English. This tilt should be taken into account especially since we correlate textual features in the corpus to readability measures that were developed - and are particularly successful - in the English language context (Antunes and Lopes, 2019).

|  | N. Titles | N. Authors |
|---|---|---|
| Whole corpus | 9089 | 7000 |
| Pulitzer | 53 | 46 |
| NBA | 104 | 79 |
| Hugo | 96 | 47 |

Table 1: Overall titles and authors in the corpus and number of long-listed titles for each award.

## 2.2 Measures of quality

We use six different measures of literary quality of two main types, heuristically setting up a qualitative distinction between more crowd-based and more expert-based measures. Expert-based measures may be supposed more institutionally prescribed, where titles are distinguished by appointing committees (as with literary prizes). Here, we chose to look at three prominent literary prizes in Anglophone literary culture: The Pulitzer Prize, the National Book Award, and the Hugo Awards, considering titles that were both long- and short-listed for these prizes. The selection of awards allows us to consider a main-stream vs. genre-literature divide in our expert measures, since the first two prizes are assigned mainly to works of literary fiction, while the latter is an award given to works of genre fiction (science fiction and fantasy).

Crowd-based measures may be considered more democratic in the sense of being user-created, for example by users' ratings on large scale reading community sites such as GoodReads, or by the effect of popular demand on library acquisitions. We use three standards here: the average ratings of titles on GoodReads (from 0 to 5 stars), the average rating count of titles on GoodReads (number of ratings given to a given title), and the number of libraries that hold a title according to Worldcat. Goodreads ratings and/or rating counts are often favoured in studies of literary quality and reception, because they seem to proffer more democratic literary evaluations "in the wild", considering the large diversity and geographical spread of its nearly 90 million users (Nakamura, 2013). In slight contrast to Goodread's ratings, we consider library holdings a conceptually hybrid measure, standing between completely free reader-based votes and expert-driven choices, as libraries respond to user-demand from within an institutional structure.

## 2.3 Measures of readability

For assessing the complexity and/or difficulty of literary texts, we apply various measures of readability. Since the 1920s, and especially with the success of the Flesch and Dale-Chall formulas in the 1950s, combinations of sentence-length and words and/or syllables have been used to assess the difficulty of a text as proxies of word and sentence complexity (Dale and Chall, 1948). According to Dubay (2004), there were more than 200 different versions of readability formulas in 1980, while new ones are still introduced and old ones revised. Still, measures from what Dubay calls the "classic" readability studies, continue to be the

---

[2]The code will be publicly available upon acceptance.
[3]While we cannot directly provide access to the corpus, it is possible to contact the authors for requests.
[4]https://www.worldcat.org/about

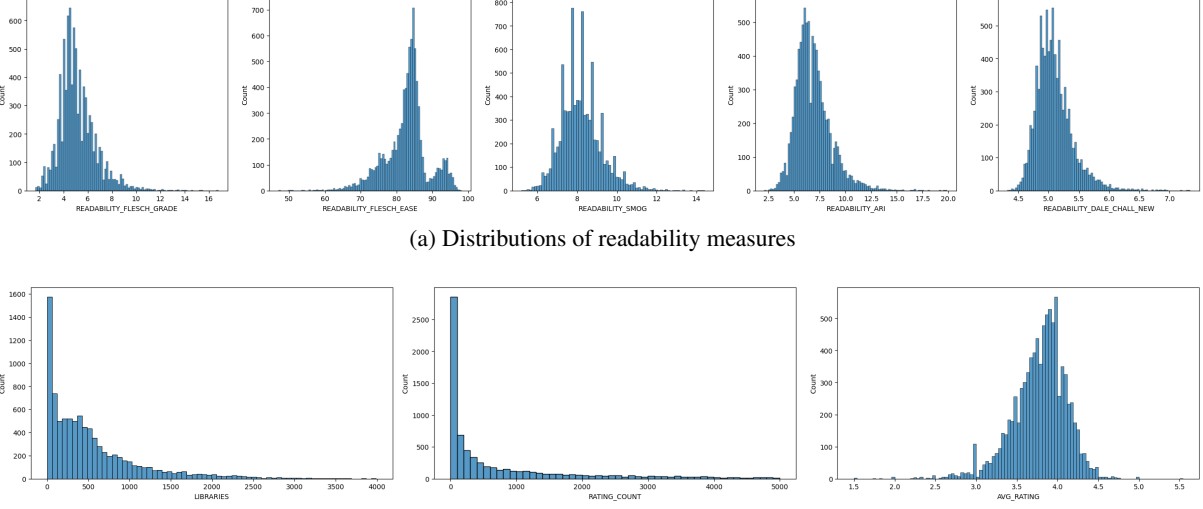

(a) Distributions of readability measures

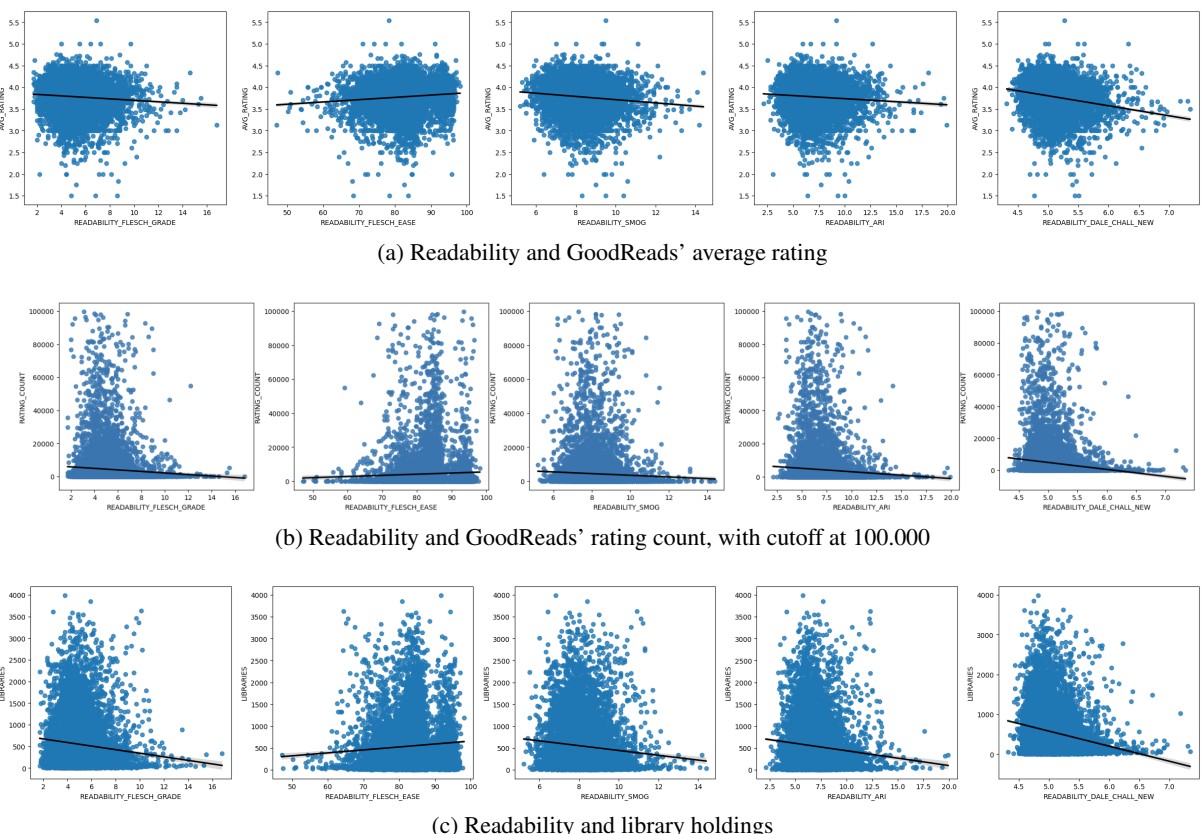

(b) Distributions of quality measures. Rating count is visualised with cutoff at 5000 for legibility.

Figure 2: Distributions of measures

(a) Readability and GoodReads' average rating

(b) Readability and GoodReads' rating count, with cutoff at 100.000

(c) Readability and library holdings

Figure 3: Quality standards and flavours of readability

most widely used measures and to prove themselves effective in assessing text difficulty (Dubay, 2004; Stajner et al., 2012) - despite their relative simplicity (being counts of two or three aspects of texts). As mentioned, readability is subjective and depends on the audience/reader. However, if the intended audience or specific reader is unkown

(as in our case), readability scores may provide a general/overarching measure which is also sufficient for comparison between texts. These measures have been applied to a wide range of written productions, from technical and journalistic texts to fiction. Flesch, for example, found that fiction tend to score a *Flesch Reading Ease* score in the

range 70 ¡ Score ¡ 90, in contrast to scientific text that often score below 30 (Flesch, 1948). In the present study we used five different "classic" readability algorithms to measure the prose of each book, chosen for their popularity and interpretability [5].

- The *Flesch Reading Ease* is a measure of readability based on the average sentence length (ASL), and the average syllables per word (word length)(ASW). It is calculated as follows:

$$\text{Score} = 206.835 - (1.015 \times \text{ASL}) - (84.6 \times \text{ASW})$$

- The *Flesch-Kincaid Grade Level* is a revised version of the Flesch Reading Ease score. Like the former, it is based on the average sentence length (ASL), and the number of syllables per word (ASW). It is calculated as follows:

$$\text{GL} = (0.4 \times \text{ASL}) + (12 \times \text{ASW}) - 15$$

- The *SMOG Readability Formula* is a readability score introduced by McLaughlin (McLaughlin, 1969). It measures readability based on the average sentence length and number of words with more than 3 syllables (number of polysyllables), applying the formula:

$$\text{SMOG grading} = 3 + \sqrt{polysyllablecount}$$

- The *Automated Readability Index* is a readability score based on the average sentence length and number of characters per words (word length). It is calculated as follows:

$$4.71 \frac{\text{characters}}{\text{words}} + 0.5 \frac{\text{words}}{\text{sentences}} - 21.43$$

- The *New Dale–Chall Readability Formula* is a 1995 revision of the Dale-Chall readability score (Chall and Dale, 1995). It is based on the average sentence length (ASL) and the percentage of "difficult words" (PDW) which were defined as words which do not appear on a list of words which 80 percent of fourth-graders would know (Dale and Chall, 1948),

contained in the Dale-Chall word-list. [6] It is calculated as follows:

$$\text{Raw Score} = 0.1579 \times \text{PDW} + 0.0496 \times \text{ASL}$$
$$\text{If PDW} > 5\% : \text{Adjusted Score} =$$
$$\text{Raw Score} + 3.6365$$

All readability scores are represented as a US-grade level, where a higher grade means a more difficult text, except for the *Flesch Reading Ease*. The *Flesch Reading Ease* indicates a score between 0 (low readability) and 100 (high readability): a higher number means a more readable text. For this reason in most of our experiments the *Flesch Reading Ease* looks reversed with respect to the other measures (and is negatively correlated with them).

## 3 Results

Pearson's and Spearman's correlations between these five readability metrics and commonly used stylometric features show - as a sanity check - that readability measures capture aspects of novels' overall style. All measures are similarly correlated to sentence-length (naturally, being a base for all measures) but also to lexical diversity and compressibility, which measure, respectively, complexity at the word- and sequence-level. Moreover, the correlations with our "quality scores" show that readability is linked with the ones closer to popularity than to appreciation.

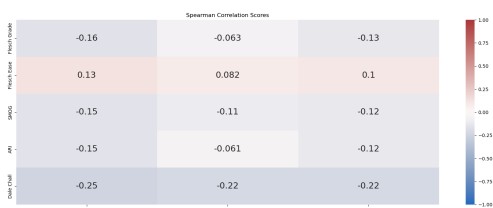

Figure 4: Correlations between quality standards and flavours of readability. All correlations are statistically significant.

Pearson's r, specifically in its significance testing, relies on the assumption of normally distributed data and it assumes that the two variables have a linear relationship, while Spearman's r correlation coefficient is non-parametric, meaning that, while it still assumes a monotonic relation

[5]All readability scores were extracted using the textstat package: https://pypi.org/project/textstat/

[6]See: https://countwordsworth.com/download /DaleChallEasyWordList.txt

between the two variables, it does not make strong assumptions on the shape of the data. For this reason, Spearman is probably the best overall measure for this study, as we have no reason to assume that all our measures are normally distributed (and some are evidently not, as can be seen in Figure 2). For these reasons, we will mainly credit the correlations observed through Spearman's r, although we report both in 2.

## 3.1 Readability and stylometrics

As readability measures are supposed to be measures of style, we compute their correlation with three core stylistic features - sentence length, lexical diversity[7] and textual compressibility[8] - that have been found linked to perceived literary quality in previous studies (van Cranenburgh and Bod, 2017; Crosbie et al., 2013; Maharjan et al., 2017; Wang et al., 2019). As can be seen in Figure 1, all readability measures have evident correlations with these three metrics, even though they don't necessarily compute them directly - for example, no readability measure computes text compressibility. However, while compressibility is not obviously correlated to readability, compressibility is a measure of redundancy or formulaicity: it appears that easier texts also have a tendency to be more sequentially repetitive. One readability measure, the new Dale-Chall, correlates with the simple length (word count) of the novels. This is a surprising effect, since, like the other measures, the new Dale-Chall is not length-dependent. As it is the only measure looking at the texts' lexicon through an index of difficult words, it seems to be picking on a tendency for longer books to have a slightly more complex vocabulary.

## 3.2 Relation with quality - GoodReads and libraries

As discussed before, we correlate readability with three possible proxies of perceived quality of novels: GoodReads' average ratings, GoodReads' rating count, and the number of libraries holding

---

[7]We operationalized lexical diversity as the type-token ratio (TTR) of a text, using a common method insensitive to text-length: the Mean Segmental Type-Token Ratio (MSTTR). MSTTR-100 represents the average TTR of local averages in 100-word segments of each text.

[8]Following van Cranenburgh and Bod (2017), for text compressibility, we calculated the compression ratio (original bit-size/compressed bit-size) using bzip2, a standard file-compressor.

a given title according to WorldCat[9]. We could

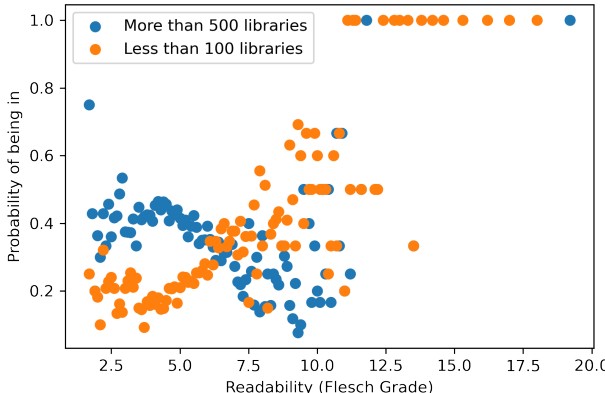

Figure 5: The likelihood of being acquired by less than 100 libraries increases quite steadily with difficulty of reading (Spearman's rho 0.84), as the probability of appearing in more than 500 declines. Readability is here measured as Flesch-Kincaid Grade Level.

consider GoodReads' rating count to be a measure closer to the concept of popularity or fame, while GoodReads' average rating tells us about the appreciation of the title independently from how many readers it had. As can be seen in Figure

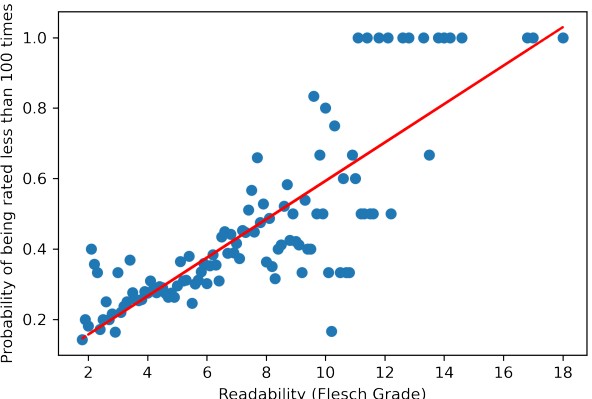

Figure 6: The probability of being rated by less than 100 users in Goodreads strongly correlates with the difficulty of the texts as measured, in this case, by the Flesch-Kincaid Grade Level.

4, all of our readability measures show a degree of correlation with the number of library holdings and the GoodReads' rating count: more readable

---

[9]Naturally this selection remains arbitrary. Expanding to other measures of perceived quality is an ongoing process.

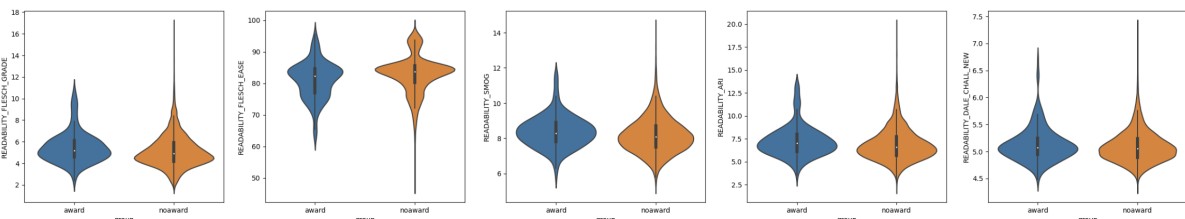

Figure 7: Flavours of readability and awards: overall distributions.

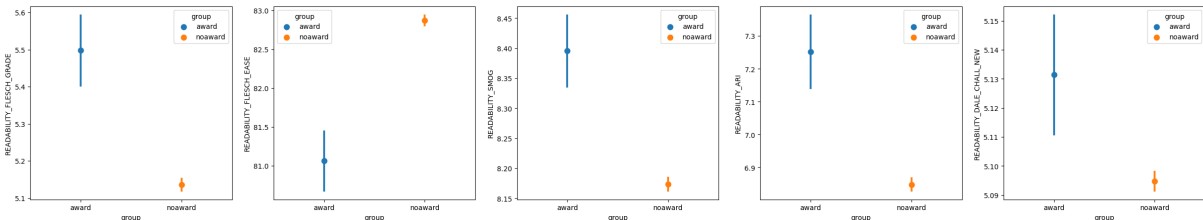

Figure 8: Flavours of readability and awards: mean value and standard error.

books tend to have more ratings and tend to be held by more libraries.

The average rating of titles on GoodReads, on the other hand, shows a significant correlation with only one of the measures, the *Dale-Chall* readability score, while it appears to have no link with the other four. Interestingly, the Dale-Chall score is the only measure that uses a precompiled list of words to estimate the number of difficult words in a text, instead of relying entirely on the features of the text at hand. While this could make it a more fragile measure (due to linguistic change and differences between genres) it appears to actually give it an increased modelling power for the tastes of GoodReads' average readers. It is worth mentioning that GoodReads' average ratings do not correlate, in our corpus, with the books' publication date - so a direct effect of language evolution on the measure's index can be excluded. Simplifying a bit, this points to the idea that the ease of vocabulary might relate to the average appreciation of a book as well as its fame, so that texts with a simpler lexicon, together with shorter sentences or words, are both more read and better liked.

In Figure 3 we show the relation of each readability measure with library holdings, average Goodreads ratings and number of Goodreads' ratings. As can be seen, we should interpret the results with some caution, as the relation might not be linear: it could be that the best interpretation of the relation between, for example, readability and library holdings is modelled with a curve rather than a straight line. Yet, it appears quite evident

at a glance that the probability of being held by a large number of libraries, and of being rated by a large number of Goodreads users, decreases dramatically when the difficulty of the text increases beyond a certain level. As we show in Figure 5, the probability of being acquired by less than 100 libraries grows quite clearly with the text's difficulty, and the probability of being acquired by more than 500 decreases accordingly, with an interesting peak at a medium-low point of difficulty. The effect is even more evident when considering the probability of having less than 100 ratings on GoodReads, as appears in Figure 6. Appearing in 90 libraries is still a quite impressive measure of success, but the majority of the titles in the Chicago corpus goes beyond that threshold, as well as beyond the threshold of 100 user ratings on GoodReads, so the difference in probabilities seems to point to a relative decline in popularity or fame with the increase of the texts' surface complexity.

### 3.3 Relation with quality - literary awards

The second type of quality check we selected is a categorical one: whether or not a title was long-listed for one of three prestigious awards - the Pulitzer Prize, the National Book Award and the Hugo Award.

As we show in Figures 7 and 8, as well as in Table 3, the difference between long-listed books and non long-listed books in terms of readability is small but significant for almost all measures, with long-listed books are systematically harder to read

|              | Libs.          | Rat. n.        |
| --- | --- | --- |
| Flesch grade | -0.16 (-0.1)   | -0.06 (-0.06)  |
| Flesch ease  | 0.13 (0.07)    | 0.08 (0.09)    |
| SMOG         | -0.15 (-0.1)   | -0.11 (-0.11)  |
| ARI          | -0.15 (-0.01)  | 0.06 (-0.06)   |
| New Dale-Chall | -0.25 (-0.2) | -0.22 (-0.2)   |
| Flesch grade | 0.84           | 0.83           |
| Flesch ease  | -0.4           | -0.48          |
| SMOG         | 0.76           | 0.81           |
| ARI          | 0.73           | 0.71           |
| New Dale-Chall | 0.78         | 0.82           |

Table 2: On the upper part of the table, Spearman's r (Pearson's in parenthesis) for each readability flavour and quality measure. On the lower, Spearman's r with the probability of being in less than 100 libraries or having less than 100 ratings.

than their non-listed counterparts - again with the exception of the new Dale-Chall measure. Using this kind of quality proxy, we do not observe a value of reading ease but possibly its "dark side", such as perceived simplification or a reduced expressive power of novels.

It may not surprise that these different standards should exhibit different preferences and perspectives on quality. Literary awards are notoriously elitist, even, perhaps, in a way that is wanted by their readership: the committee of the Booker Prize was accused of populism in 2011 when announcing "readability" as a new criterion for the award (Clark, 2011).

|              | T-test | p-value  |
| --- | --- | --- |
| Flesch grade | 3.78   | 0.0001   |
| Flesch ease  | -4.66  | 0.000005 |
| SMOG         | 3.69   | 0.0002   |
| ARI          | 3.6    | 0.0003   |
| New Dale-Chall | 1.8  | 0.07     |

Table 3: T-test and p-value for the difference between long-listed and non-listed titles for each readability measure. The only measure that does not fall under the formal threshold of statistical significance is the new Dale-Chall.

## 4 Conclusions and Future Works

Readability measures proved significantly consistent, both between each other and with other relevant stylometric features, when applied on modern and contemporary fiction. Their relation with different proxies of literary quality is intriguing: more popular works, in terms of number of ratings on GoodReads and in terms of libraries willing to hold a copy of the book, appear to have a correlation with readability, while the appreciation of readers alone (independently from their number) seems to hold almost no link with it, and long-listed titles have an inverse relation with readability, tending to prefer slightly more difficult prose on the readability metrics' scale. It can be argued that we are seeing the divide between high-brow and "popular" literature, but the lack of correlation with GoodReads average rating might point to a slightly more nuanced conclusion. It is worth noting that the only measure showing a meaningful correlation with all of the crowd-based quality metrics was the new Dale-Chall measure of readability, also the only one explicitly focusing on the presence of widely understood lexicon in a text, but it was also the only one showing no significant difference between long-listed and non long-listed titles. The only other measure having a correlation higher than 0.1 with average GoodReads' ratings was SMOG, which, while not using a list of hard words, considers "difficult words" in its own way in its computation, using the number of polysyllable words as a central element. If we were to draw rough conclusions from these observations, it would seem that surface-level simplicity of style in terms of words per sentence, characters per words, and similar metrics "helps" a text's popularity, but has nothing to do with its likelihood of being highly liked by its readers - and it even slightly hinders its possibilities of receiving a prestigious awards. In other words, surface-level simplicity improves a text's quality only if we equate it with popularity or fame. Similarly, looking at threshold-based probability distributions showed that indeed increasing the difficulty of the novels' style might hinder its diffusion across libraries and Goodreads' users. Using a more common vocabulary might also increase readers' appreciation of the text, but only when it comes to crowd-based measures. On the other hand, the correlations of average number of ratings and library holdings with readability measures do not appear linear or monotonic, meaning that there might also be a "point of balance" between too easy and too difficult, that maximizes the correlation with a novel's fame. The same might be true for the likelihood

of a novel being long-listed for one of the three awards we took into consideration.

Overall, readability seems to have an impact on different perceptions of literary quality, although its role and interaction with other features of the text remains to be defined. Another overarching point to observe from these findings is that there is a difference between crowd-based (GoodReads) and expert-based (awards) standards of literary quality in readability-level preference, which indicates that the criteria change across different quality-judgements, which suggetss that "literary quality" cannot be quantified reliably if it is reduced to a single golden standard. Further research points towards extending the set of correlations to more proxies of quality as well as more sophisticated stylometric measures to see whether interactions can provide a clearer picture of what we perceive as literary quality. Other further work could be to check the correlations of our measures with publication date: readability might depend on time, either in the sense of the evolution of the average novelistic style, overall language change, or even cultural selection, which would make the passage of time a particular form of "quality test" of its own accord.

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
