# OpenReview forum: "Good Reads and Easy Novels: Readability and Literary Quality in a Corpus of US-published Fiction"
_NoDaLiDa/2023/Conference — NoDaLiDa 2023_

### Official Review · Reviewer_ExEN · 2023-03-09
**Strong paper with interesting findings**

**Rating:** 7
**Confidence:** 3

**Review:**

The authors examine the relationship between the concept of readability and the literary quality of texts, both from an expert- and casual-reader-based perspective.

The reported work is thoroughly carried out and documented, with clear writing, good background surveys, and a well-established methodology. One drawback to the study and the formulae used within is the anglocentricity of the content and methods, which the authors do acknowledge. It would be interesting to include a note, either in the conclusion or while discussing various methods, on their applicability to or similar work for other languages, or even lack thereof, depending on what the authors know from their background research.

Regarding the Flesch Reading Ease score (line 477 onwards), would it be helpful to include an inverse/negative Flesch in Figure 3, for ease of graph interpretation? (pun not intended)

Finally, Figures 5, 7 and 8 are difficult to interpret in greyscale -- perhaps update the figures with a colour palette that is both colourblind- and greyscale-friendly?


Typos and minor issues:

020: repetition in abstract

084: repetition of "often"

497: between with our?

Pearson's and Spearman's typos: 512, 515, 526

668: odd formatting artefact

824: singular or plural award?



**Paper Type:**

Long paper

---

### Official Review · Reviewer_yWWE · 2023-03-14
**Interesting Read**

**Rating:** 7
**Confidence:** 3

**Review:**

The authors explore the relationship between (multiple measures of) readability and (multiple measures of) success of a novel.
Their findings suggest that higher readability (i.e., being easier to read) helps a book's popularity, but is inversely correlated with its chances of winning a price.

The paper is well written, the experiments seem to have been conducted well, and the findings are interesting. Thus I'd be in favor of seeing it accepted.

**Paper Type:**

Long paper

---

### Decision · Program_Chairs · 2023-03-17

Accept